

# Selecting allometric equations to estimate forest biomass from plot- rather than individual-level predictive performance

Nicolas Picard[1], Noël Fonton[2], Faustin Boyemba Bosela[3], Adeline Fayolle[4,5], Joël Loumeto[6], Gabriel Ngua Ayecaba[7], Bonaventure Sonké[8], Olga Diane Yongo Bombo[9], Hervé Martial Maïdou[10], and Alfred Ngomanda[11]

[1]GIP Ecofor, Paris, France
[2]University of Abomey-Calavi, Cotonou, Benin
[3]University of Kisangani, Kisangani, Democratic Republic of Congo
[4]Forêts et Sociétés, Université de Montpellier, Cirad Montpellier, France
[5]Cirad Forêts et Sociétés, Montpellier, France
[6]University Marien NGouabi, Brazzaville, Republic of Congo
[7]Instituto Nacional de Desarrollo Forestal y Manejo del Sistema Nacional de Areas Protegidas (INDEFOR), Bata, Equatorial Guinea
[8]University of Yaoundé 1, Yaounde, Cameroon
[9]University of Bangui, Bangui, Central African Republic
[10]Commission des Forêts d'Afrique Centrale (COMIFAC), Yaounde, Cameroon
[11]Centre National de la Recherche Scientifique et Technologique (CENAREST), Libreville, Gabon

**Correspondence:** Nicolas Picard (nicolas.picard@gip-ecofor.org)

**Abstract.** In a context of global change, it is essential to quantify and monitor the carbon stored in forests. Allometric equations are mathematical models that predict the biomass of a tree from dendrometrical characteristics that are easier to measure, such as tree diameter, height or wood density. Various model forms have been proposed for allometric equations. Moreover, the model choice has a critical influence on the estimate of the biomass of a forest. So far, model selection for allometric

equations has been performed based on the tree-level predictive performance of the models. Yet, allometric equations are used to estimate the biomass of plots rather than individual trees. The distribution of trees sampled for establishing allometric equations often differs from the forest structure. Moreover, at the plot-level, the residual individual errors for different trees can cancel off. Therefore, we expect the plot-level predictive performance of a model to differ from its tree-level performance. Using a dataset giving the observed biomass of 844 trees in central Africa and a null model for the size distribution of trees

in the forest, we simulated forest plots between 0.1 and 50 ha in area. Then, using a Monte Carlo approach, we calculated the mean sum of squares (MSS) of the differences between observed and predicted plot biomass. We showed that MSS could be well approximated by a three-term formula, where the first term corresponded to bias, the second one to the tree residual error, and the third one to the uncertainty on model coefficients. For small plots ($\leq 0.1$ ha), the plot-level predictive performance was dominated by the tree residual error term. Model selection based on plot-level predictive performance was then consistent

with that based on tree-level performance. For large plots, this term vanished. Model selection based on plot-level performance could then differ from that based on tree-level performance. In the case of large plots, chains of models that combined a general equation to predict biomass and local equations to predicts some of the predictors of the biomass equation could





provide a good trade-off between the bias and the uncertainty on model coefficients. We recommend using plot-level rather than tree-level predictive performance to select allometric equations. The three-term formula that we developed provides an easy way to assess the effect of plot size on model selection and to balance the respective contributions of bias, tree residual error, and the uncertainty on model coefficients.

## 1 Introduction

In a context of changing climate due to increasing $CO_2$ atmospheric concentration, it is essential to quantify and monitor the main compartments that store or emit carbon at the global level. Forests are one of these compartments and are part of the solutions to mitigate climate change (Lewis et al., 2019). Measuring and monitoring forest carbon stocks involves a chain of measurements that starts with biomass measurement at individual tree level and ends with remote sensing techniques (Gibbs et al., 2007; Réjou-Méchain et al., 2019). Typically, tree-level biomass measurements are used to fit allometric equations that predict the biomass of a tree from tree dendrometrical characteristics that are easier to measure, such as diameter, height or wood density (Chave et al., 2014). Allometric equations are used in turn to estimate the biomass of forest plots. Plot-level biomass data are used to calibrate remote sensing indices to predict the biomass of pixels in satellite images. Satellite images are finally used to map forest biomass on large areas. In this study, we focus on the step of the allometric equation that connects the tree level to the plot level.

Despite the proposal to skip the step of allometric equations by directly measuring biomass at the plot level (Clark and Kellner, 2012), allometric equations remain an indispensable link in the measurement chain (Vorster et al., 2020). Fast-developing measurement techniques like terrestrial or airborne LiDAR may provide in the future plot-level measures of biomass (Xu et al., 2021). These same techniques give a greater level of details in the description of trees (Momo Takoudjou et al., 2018; Lines et al., 2022). They thus also give the opportunity to integrate new predictors of tree biomass in allometric equations, such as crown dimensions, trunk shape, diameter of the largest branches, or tree architecture (MacFarlane, 2011; Goodman et al., 2014; Brede et al., 2022). So far, the use of predictors in allometric equations has been mostly limited by the set of dendrometrical variables commonly available in forest inventories. In tropical forests, these variables are usually limited to diameter and species. Adding relevant tree-level predictors would provide a better understanding of individual tree variability in biomass and reduce the residual standard error of allometric equations. Additional predictors may also reduce the prediction bias and increase overall the predictive performance of allometric equations at the tree level.

Yet, allometric equations are intended to provide biomass estimates at the plot rather than the tree level (McRoberts and Westfall, 2014; McRoberts et al., 2015; Paul et al., 2016). Therefore, their predictive performance should be assessed at the plot rather than at the individual level. The comparison and selection of allometric equations is commonly based on the goodness-of-fit of the fitted models, using selection criteria like the Akaike information criterion (AIC), the Bayesian information criterion (BIC) or the root mean square error (RMSE). This selection mode puts the emphasis on the predictive performance of the models at the tree level. Nevertheless, a large residual error at the tree level may be compensated at the plot level when the residual errors from different trees cancel off. The levelling off of the individual residual error is all the more important as the



plot is large. Thus, explaining the greatest share of the variance in tree biomass may not always be the best strategy to select an allometric equation to predict plot biomass. Moreover, adding dendrometrical predictors to the model to reduce the tree-level residual error will inflate the measurement cost to get these additional predictors at the plot level.

From a statistical standpoint, criteria like AIC or BIC may be tricky to use to compare models that have been fitted in different ways (e.g. ordinary-least-squares fitting on log-transformed data versus weighted-least-squares fitting on untransformed data). Another difficulty is to compare chains of models, e.g. a model that predicts $y$ from $x$ followed by another model that predicts tree biomass from $y$, versus a third model that directly predicts tree biomass from $x$. Such a situation is often found when it comes to the role of tree height in the prediction of its biomass. Tree height generally improves the prediction of biomass but is rarely available in large scale forest inventories (Chave et al., 2014). On the other hand, datasets on tree height are much more abundant than datasets on tree biomass, so that a diameter-height model can usually be fitted with higher precision than biomass models (Feldpausch et al., 2011). Thus, one option is to predict height from diameter, then biomass from diameter and height (Feldpausch et al., 2012). Another option is to predict biomass from diameter alone. A pending question that cannot be addressed using AIC or BIC is which option is the best.

We here proposed a method to compare allometric equations that focused on the model predictive performance at the plot level rather than at the tree level. Different competing models were thus compared. We placed ourselves in the context of fitting allometric equations, when a calibration dataset of observed tree biomass is available and model coefficients need to be estimated. A different context is when allometric equations are given with known coefficients, and a validation dataset is given to compare their predictive performance. The method we proposed can accommodate models fitted in different ways. It can also be used to compare chains of models. It is a Monte Carlo method that relied on randomly generated plot-level data, thus allowing to compare equations for different plot sizes. Given a dataset on individual tree attributes (including tree biomass), a plot was generated by randomly picking trees while constraining plot structural characteristics (such as tree density or basal area) to prescribed values. These plot structural characteristics were set using a null forest model. We used a dataset on tree biomass in the Congo basin to illustrate the method (Fayolle et al., 2018).

Using this dataset, we addressed the following questions: (*i*) Does model selection based on predictive performance at tree level (e.g. using AIC, BIC or RMSE) agree with model selection based on predictive performance at plot level? (*ii*) How does plot size affect model selection when this selection is based on predictive performance at plot level? (*iii*) When extra data on tree height is available so that a height-diameter model can be fitted, does predicted height improve the prediction of biomass through a chain of models? We hypothesized that the role of the residual model error, that is decisive in tree-level predictive performance, is decreasing with plot size when evaluating plot-level predictive performance.

## 2 Material and methods

The predictive performance of allometric equations was assessed (*i*) at the tree level based on a dataset of tree biomass observations, (*ii*) at the plot level based on randomly generated plots using a null forest model and the tree dataset, and (*iii*) at the





**Table 1.** Statistics used to assess the predictive performance of allometric models at the tree, plot and forest levels. $A$ is the plot area and $N$ is the tree density.

| Level | Statistics |
|---|---|
| Tree | $b_\mathcal{X}$, $\text{MSE}_\mathcal{X}$, $\text{ME}_\mathcal{X}$, AIC, $\sigma$, $R^2$ |
| Forest | $b_\mathcal{F}$, $\text{MSE}_\mathcal{F}$, $\text{ME}_\mathcal{F}$ |
| Plot | $(Nb_\mathcal{F})^2$, $(N/A)\text{MSE}_\mathcal{F}$, $N^2\text{ME}_\mathcal{F}$, MSS |

forest level based on the same null forest model. For each of these three levels, specific performance statistics were used (Table 1).

## 2.1 Tree biomass data and tree-level predictive performance

We used the dataset on individual tree biomass described in Fayolle et al. (2018). This dataset, denoted $\mathcal{X}$, includes the diameter, height, wood specific gravity, aboveground biomass and species of $m = 844$ trees in the Congo basin. Trees belong to 52 different species and 49 different genera. Data were collected in six countries of the Congo basin: Cameroun, Central African Republic, Congo, Democratic Republic of Congo, Equatorial Guinea and Gabon. Details on tree measurements and data collection are given in Fayolle et al. (2018). Trees in dataset $\mathcal{X}$ were sampled in the range 10.3–208.0 cm in dbh, with a peak of the sampling effort around 35 cm dbh (Fig. 1). Let $d_\mathcal{X}$ be the density of the diameter distribution of trees in dataset $\mathcal{X}$. This distribution reflects the sampling design of trees and is unrelated to the diameter distribution of trees in the forest.

Let $f$ be an allometric equation that predicts the tree biomass $f(\mathbf{x}, \theta)$ of each tree using its dendrometrical characteristics $\mathbf{x}$, where $\theta$ denotes the coefficients of the model. All the models we considered were fitted using linear regression on log-transformed data. Even when using a bias-correction factor, the back-transformation introduced a prediction bias. Let $B_i$ the observed biomass of the $i$th tree of $\mathcal{X}$. Thus, the bias of prediction of the biomass of a randomly selected tree in dataset $\mathcal{X}$ is: $b_\mathcal{X} = \frac{1}{m} \sum_{i=1}^{m} [B_i - f(\mathbf{x}_i, \theta)]$. The mean square error for a tree of dataset $\mathcal{X}$ is: $\text{MSE}_\mathcal{X} = \frac{1}{m} \sum_{i=1}^{m} [B_i - f(\mathbf{x}_i, \theta)]^2$. Although rarely considered in the statistics of predictive performance of allometric equations, one may also consider the prediction variability brought by the uncertainty on the model coefficients $\theta$. When using a linear regression to fit the model, the estimator of $\theta$ is distributed as a multivariate normal distribution with mean $\theta$ and covariance matrix $\Sigma$. Therefore, one may use the following mean error to assess the uncertainty on the model coefficients: $\text{ME}_\mathcal{X} = \int_\vartheta \{\frac{1}{m} \sum_{i=1}^{m} [f(\mathbf{x}_i, \theta) - f(\mathbf{x}_i, \vartheta)]\}^2 \, \Phi(\vartheta, \theta, \Sigma) \, \mathrm{d}\vartheta$, where $\Phi(\cdot, \theta, \Sigma)$ is the density of the multivariate normal distribution with mean $\theta$ and covariance matrix $\Sigma$.

As a secondary dataset, denoted $\mathcal{X}'$, we used a subset of the pantropical dataset assembled before $\mathcal{X}$ by Chave et al. (2014). We kept only observations from the Congo basin (Cameroon, Central African Republic and Gabon), totaling $m' = 177$ trees. The dataset gives the diameter, height, wood specific gravity and aboveground biomass of trees. However, for the purposes of our study, we only kept the diameter and height variables.





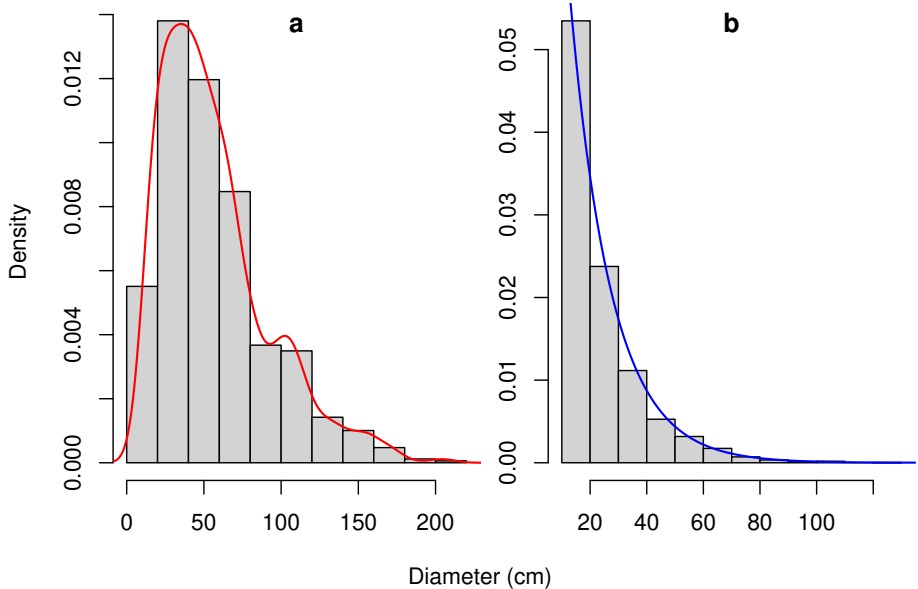

**Figure 1.** Diameter distribution of (a) the 844 trees sampled in the Congo basin forests for the measurement of their biomass (Fayolle et al., 2018) and (b) 10,000 trees resampled from the former set of trees so as to conform to an exponential distribution with parameter 0.0689 cm$^{-1}$. The red line is a density estimate $d_{\mathcal{X}}$ of the distribution using a Gaussian kernel with a bandwidth determined by Silverman's rule-of-thumb. The blue line is the density $d_{\mathcal{F}}$ of the exponential distribution with parameter 0.0689 cm$^{-1}$. The dataset in (b) was obtained from the dataset in (a) by resampling each diameter $x$ with a probability proportional to $d_{\mathcal{F}}(x)/d_{\mathcal{X}}(x)$.

## 2.2 Null forest stand model and forest-level predictive performance

Plot-level biomass data were generated using the collection of tree biomass measurements and a null model for the diameter structure of the forest. This null model had two entries: the stand density $N$ and its basal area $G$. Following the hypothesis of demographic equilibrium, the null model assumed that the forest had a reverse-J shaped diameter distribution that could be modeled by an exponential distribution (Muller-Landau et al., 2006; Picard et al., 2021). The parameter $\mu$ of this exponential distribution can be computed from $N$ and $G$ as: $\mu = [\sqrt{2G/(\pi N) - x_0^2/4} - x_0/2]^{-1}$, where $x_0$ is the minimum diameter for inventory in the forest plot. We used for $N$ and $G$ average values given by Picard et al. (2021), based on sample plots in central Africa: $N = 467$ ha$^{-1}$ and $G = 29.8$ m$^2$ ha$^{-1}$. These values gave: $\mu = 0.0689$ cm$^{-1}$.

An outcome $\mathcal{Y}$ of the null forest was randomly drawn by resampling dataset $\mathcal{X}$ so that the diameter distribution of trees in $\mathcal{Y}$ conformed to the exponential distribution with parameter $\mu$. Let $d_{\mathcal{F}}(x) = \mu \exp[-\mu(x - x_0)]$ be the density of this target distribution. Because the diameter distibution of trees in $\mathcal{X}$ differed for the target distribution, the resampling involved unequal weights. Specifically, the $i$th trees of $\mathcal{X}$ was resampled with a weight $w_i$ proportional to $d_{\mathcal{F}}(x_i)/d_{\mathcal{X}}(x_i)$. In other words,

$$w_i = \frac{d_{\mathcal{F}}(x_i)/d_{\mathcal{X}}(x_i)}{\sum_{j=1}^{m} d_{\mathcal{F}}(x_j)/d_{\mathcal{X}}(x_j)} \tag{1}$$





so that $\sum_{i=1}^{m} w_i = 1$. For a forest plot with area $A$, the $N \times A$ trees in $\mathcal{Y}$ were thus sampled from $\mathcal{X}$ with replacement using the probability of drawing $w_i$ for the $i$th tree of $\mathcal{X}$. For an allometric equation $f$, the bias of prediction of the biomass of a randomly chosen tree in the forest according to the null model was therefore:

$$\mathrm{b}_{\mathcal{F}} = \sum_{i=1}^{m} w_i [B_i - f(\mathbf{x}_i, \theta)] \tag{2}$$

The mean square error for a randomly chosen tree in the forest was:

$$\mathrm{MSE}_{\mathcal{F}} = \sum_{i=1}^{m} w_i [B_i - f(\mathbf{x}_i, \theta)]^2 \tag{3}$$

As for the uncertainty on the model coefficients, it was assessed for a randomly chosen tree in the forest by the following mean error:

$$\mathrm{ME}_{\mathcal{F}} = \int_{\vartheta} \left\{ \sum_{i=1}^{m} w_i [f(\mathbf{x}_i, \theta) - f(\mathbf{x}_i, \vartheta)] \right\}^2 \Phi(\vartheta, \theta, \Sigma) \, \mathrm{d}\vartheta \tag{4}$$

### 2.3 Error partitioning and plot-level predictive performance

The different sources of prediction error at the plot level were assessed using a Monte Carlo approach. Plot variability in biomass predictions was generated by drawing different outcomes of the null forest stand model. The variability due to model coefficients was generated by drawing different outcomes of the model coefficients according to a multivariate normal distribution with mean $\theta$ and covariance matrix $\Sigma$. Calculations were done by combining each plot outcome with each coefficient outcome, resulting in a full factorial design.

#### 2.3.1 For a model

Let $K$ be the number of randomly generated forest plots and let $J$ be the number of randomly generated model coefficients. Let $n_k$ be the number of trees in the $k$th plot, let $\theta_j$ be the $j$th outcome of the model coefficients, and let $\mathbf{x}_{ki}$ and $b_{ki}$ be the dendrometrical characteristics and observed biomass of the $i$th tree of the $k$th plot. Let $e_{kj} = [\sum_{i=1}^{n_k} b_{ki} - f(\mathbf{x}_{ki}, \theta_j)]/A$ be the difference between the observed biomass of the $k$th plot and its predicted biomass according to model $f$ using the $j$th coefficient value, per unit of plot area. The plot-level predictive performance of model $f$ was assessed using the mean sum of squares of these differences:

$$\mathrm{MSS} = \frac{1}{KJ} \sum_{k=1}^{K} \sum_{j=1}^{J} e_{kj}^2 \tag{5}$$

By the definition of the variance, the mean sum of squares is equal to the variance plus the squared bias. As the plot area tends to infinity, the number of trees goes to infinity and the difference between the plot-level observed biomass and the predicted





one tends towards the forest-level bias times the number of trees in the plot. Thus: MSS = Var + SB, where:

$$\text{SB} \quad = \quad \bar{e}^2 = \left( \frac{1}{KJ} \sum_{k=1}^{K} \sum_{j=1}^{J} e_{kj} \right)^2 \approx (N\mathbf{b}_{\mathcal{F}})^2 \tag{6}$$

$$\text{Var} \quad = \quad \frac{1}{KJ} \sum_{k=1}^{K} \sum_{j=1}^{J} (e_{kj} - \bar{e})^2 \tag{7}$$

Using the calculations of the analysis of variance, the variance can in turn be partitioned into an inter-plot variance (or plot variability) and an intra-plot variance. The variability in plot-level biomass errors results from the individual tree errors that do not compensate. Therefore, the greater the residual standard error of model $f$, the greater this plot variability. As the plot area tends to infinity, the tree-level errors from different trees compensate each other and the plot variability vanishes. As the plot area tends to zero, the tree-level errors from different trees do not compensate. For a very small plot area, the plot contains very few trees whose errors accumulate almost independently. Therefore, the plot-level variance of biomass differences is close to the sum of individual errors: $NA \times \text{MSE}_{\mathcal{F}}$. Scaling this error per unit area of the plot finally brings: $(N/A)\,\text{MSE}_{\mathcal{F}}$.

As regards the intra-plot variance (or variability within a plot), it results from the different coefficient values and reflects the uncertainty on these coefficients. For a tree taken at random in the forest with probability $w_i$, the difference in biomass due to a model coefficient $\theta_j$ is: $\sum_{i=1}^{m} w_i[f(\mathbf{x}_i, \theta) - f(\mathbf{x}_i, \theta_j)]$. For the $NA$ trees found in a plot with area $A$, this difference is multiplied by $NA$. Integrating over the possible outcomes of $\theta_j$ and scaling per unit area of the plot, it shows that the coefficient variability is close to: $N^2\,\text{ME}_{\mathcal{F}}$.

To summarize, Var = (plot variability) + (coefficient variability), where:

$$\text{plot variability} \quad = \quad \frac{1}{K} \sum_{k=1}^{K} (\bar{e}_{k.} - \bar{e})^2 \approx (N/A)\,\text{MSE}_{\mathcal{F}} \tag{8}$$

$$\text{coefficient variability} \quad = \quad \frac{1}{KJ} \sum_{k=1}^{K} \sum_{j=1}^{J} (e_{kj} - \bar{e}_{k.})^2 \approx N^2\,\text{ME}_{\mathcal{F}} \tag{9}$$

$$\bar{e}_{k.} \quad = \quad \frac{1}{J} \sum_{j=1}^{J} e_{kj} \tag{10}$$

### 2.3.2 For a chain of models

We generalized the assessment of the predictive performance of an equation to a chain of two allometric equations, where the response variable of the first equation is a predictor of the second one. Our computations readily extends to a chain of three or more allometric equations. Let $g$ be an allometric equation that predicts some tree characteristics $\mathbf{y} = g(\mathbf{x}, \phi)$ from some dendrometrical characteristics $\mathbf{x}$ of the tree. Let $f$ be a second allometric equation that predicts the tree biomass $f(\mathbf{x}, \mathbf{y}, \theta)$ using its dendrometrical characteristics $\mathbf{x}$ and those predicted by model $g$. Coefficients $\theta$ and $\phi$ are those of models $f$ and $g$, respectively.

To the $K$ randomly generated plots and $J$ randomly drawn coefficients $\theta$, we now add $L$ random draws of the coefficients $\phi$. Let $\phi_l$ be the $l$th outcome of the model coefficients. Let $e_{klj} = \{\sum_{i=1}^{n_k} b_{ki} - f[\mathbf{x}_{ki}, g(\mathbf{x}_{ki}, \phi_l), \theta_j]\}/A$ be the difference between





the observed biomass of the $k$th plot and its predicted biomass according to the chain $f \circ g$ using the $j$th coefficient value of $f$ and the $l$th coefficient value of $g$, per unit of plot area. The mean sum of squares of these differences now is:

$$\text{MSS} = \frac{1}{KLJ} \sum_{k=1}^{K} \sum_{l=1}^{L} \sum_{j=1}^{J} e_{klj}^2 \tag{11}$$

As before, the mean sum of squares is the variance plus the squared bias: MSS = Var + SB, where:

$$\text{SB} = \bar{e}^2 = \left( \frac{1}{KLJ} \sum_{k=1}^{K} \sum_{l=1}^{L} \sum_{j=1}^{J} e_{klj} \right)^2 \tag{12}$$

$$\text{Var} = \frac{1}{KLJ} \sum_{k=1}^{K} \sum_{l=1}^{L} \sum_{j=1}^{J} (e_{klj} - \bar{e})^2 \tag{13}$$

Again, the variance can be partitioned into an inter-plot variance (or plot variability) and an intra-plot variance (or coefficient variability): Var = (plot variability) + (coefficient variability), where:

$$\text{plot variability} = \frac{1}{K} \sum_{k=1}^{K} (\bar{e}_{k..} - \bar{e})^2 \tag{14}$$

$$\text{coefficient variability} = \frac{1}{KLJ} \sum_{k=1}^{K} \sum_{l=1}^{L} \sum_{j=1}^{J} (e_{klj} - \bar{e}_{k..})^2 \tag{15}$$

$$\bar{e}_{k..} = \frac{1}{LJ} \sum_{l=1}^{L} \sum_{j=1}^{J} e_{klj} \tag{16}$$

Now, the coefficient variability can be partitioned into the variability due to the coefficients of model $g$ and that due to the coefficients of model $f$: (coefficient variability) = (variability due to $g$ coefficients) + (variability due to $g$ coefficients), where:

$$\text{variability due to } g \text{ coefficients} = \frac{1}{KL} \sum_{k=1}^{K} \sum_{l=1}^{L} (\bar{e}_{kl.} - \bar{e}_{k..})^2 \tag{17}$$

$$\text{variability due to } f \text{ coefficients} = \frac{1}{KLJ} \sum_{k=1}^{K} \sum_{l=1}^{L} \sum_{j=1}^{J} (e_{klj} - \bar{e}_{kl.})^2 \tag{18}$$

$$\bar{e}_{kl.} = \frac{1}{J} \sum_{j=1}^{J} e_{klj} \tag{19}$$

## 2.4 Model comparisons

We compared five allometric equations and one chain of equations. The five models were:

$$\ln(B) = a_1 + b_1 \ln(\rho D^2 H) \tag{20}$$

$$\ln(B) = a_{2s} + b_{2s} \ln(\rho D^2 H) \tag{21}$$

$$\ln(B) = a_3 + b_3 \ln(\rho) + c_3 \ln(D) \tag{22}$$

$$\ln(B) = a_4 + b_4 \ln(\rho) + c_4 \ln(D) + d_4 \ln(H) \tag{23}$$

$$\ln(B) = a_5 + b_5 \ln(\rho) + c_5 \ln(D) + d_5 \ln(H) + e_5 [\ln(D)]^2 \tag{24}$$





**Table 2.** Statistics on the predictive performance of five allometric equations fitted to a dataset $\mathcal{X}$ of 844 trees in the Congo basin forests. The response variable of all these models is the log-transformed tree aboveground biomass. $N$ is the density of a forest plot with area $A = 1$ ha, distributed according to a null forest model $\mathcal{F}$. The quantity b is the prediction biases of tree biomass; MSE is the mean square error of tree biomass; ME is the mean error due to coefficient uncertainty. For these three quantities, subscripts $\mathcal{F}$ and $\mathcal{X}$ refer to the null forest and to the fitting dataset. AIC is Akaike Information Criterion. $\sigma$ is the residual standard error and $R^2$ is the coefficient of determination of the fitted model.

| Model | $(Nb_{\mathcal{F}})^2$ | $(Nb_{\mathcal{X}})^2$ | $(N/A)$ $\mathrm{MSE}_{\mathcal{F}}$ | $(N/A)$ $\mathrm{MSE}_{\mathcal{X}}$ | $N^2 \times$ $\mathrm{ME}_{\mathcal{F}}$ | $N^2 \times$ $\mathrm{ME}_{\mathcal{X}}$ | AIC | $\sigma$ | $R^2$ |
|---|---|---|---|---|---|---|---|---|---|
| (20) $a_1 + b_1 \ln(\rho D^2 H)$ | 22.48 | 76.52 | 63.97 | 3 370.0 | 6.87 | 1 068.6 | 91.8 | 0.255 | 0.98 |
| (21) $a_{2s} + b_{2s} \ln(\rho D^2 H)$ | 24.89 | 31.95 | 36.76 | 1 722.7 | 5.67 | 879.1 | −164.4 | 0.208 | 0.99 |
| (22) $a_3 + b_3 \ln(\rho) + c_3 \ln(D)$ | 2.50 | 17 729.3 | 66.97 | 4 622.1 | 7.98 | 1 424.2 | 173.8 | 0.267 | 0.97 |
| (23) $a_4 + b_4 \ln(\rho) + c_4 \ln(D) + d_4 \ln(H)$ | 0.19 | 1 804.4 | 59.35 | 3 439.5 | 6.90 | 1 099.7 | 24.0 | 0.245 | 0.98 |
| (24) $a_5 + b_5 \ln(\rho) + c_5 \ln(D) + d_5 \ln(H)$ $+ e_5 [\ln(D)]^2$ | 0.06 | 2 540.9 | 59.72 | 3 503.8 | 7.09 | 1 430.8 | 25.8 | 0.245 | 0.98 |

where $\rho$ was the wood specific gravity in $\mathrm{g\,cm}^{-3}$, $D$ was tree diameter in cm, $H$ was tree height in m, and $s$ denoted the tree genus. We used F-tests to compare nested models, i.e. to compare models (20) and (21), models (22) and (23), and models (23) and (24). In the chain of equations, the first model predicted tree height from tree diameter:

$$\ln(H) = a_6 + b_6 \ln(D) \tag{25}$$

while the second model was model (20).

Models (20)–(24) were fitted to dataset $\mathcal{X}$ with $m = 844$ observations, while model model (25) was fitted to $\mathcal{X} \cup \mathcal{X}'$ with $m + m' = 1021$ observations. When back-transforming the data from the log-transform, the bias correction factor $\exp(\sigma^2/2)$ was used, where $\sigma$ was the residual error of the fitted model. Monte Carlo computations were done with $K = 1000$ and $J = 1000$, bringing $10^6$ values of $e_{kj}$. For the chain assessment, we used $K = 800$ and $J = L = 50$, bringing $2 \cdot 10^6$ values of $e_{klj}$. All computations were done with the R software.

## 3 Results

When looking at tree-level performance statistics, the best model was model (21). It had at the same time the lowest AIC, the smallest residual standard error $\sigma$, the smallest prediction bias $Nb_{\mathcal{X}}$, the smallest mean square error $(N/A)\,\mathrm{MSE}_{\mathcal{X}}$, and the smallest mean error due to coefficient uncertainty $N^2\,\mathrm{ME}_{\mathcal{X}}$ among the five competing models (Table 2). When looking at forest-level performance statistics, model (21) still had the smallest mean square error $(N/A)\,\mathrm{MSE}_{\mathcal{F}}$ and the smallest mean error due to coefficient uncertainty $N^2\,\mathrm{ME}_{\mathcal{F}}$ among the five models (Table 2). However, it was the least performing model in terms of prediction bias $Nb_{\mathcal{F}}$. The model with the smallest forest-level prediction bias $Nb_{\mathcal{F}}$ was model (24).





When looking at the partition of MSS, the forest-level squared biased $(Nb_{\mathcal{F}})^2$ was a good approximation of the squared bias (SB) component of MSS for plot area greater than 50 ha (Fig. 2c). The SB component of MSS actually showed little fluctuations around $(Nb_{\mathcal{F}})^2$ as plot area changed (Fig. 2c). The forest-level mean error due to coefficient uncertainty $N^2 ME_{\mathcal{F}}$ was also a good approximation of the coefficient variability component of MSS for plot area greater than 50 ha (Fig. 2a). Like SB, the coefficient variability showed little fluctuations around $N^2 ME_{\mathcal{F}}$ as plot area changed (Fig. 2a). In contrast, the plot variability component of MSS sharply decreased with plot area (Fig. 2b). It actually decreased proportionally to the inverse of plot area, with the coefficient of proportionality being well approximated by $N MSE_{\mathcal{F}}$.

Adding up the various components of MSS, the predictive performance of a model for the biomass of a forest plot thus depended on plot area. For a small plot area of 0.1 ha, MSS was dominated by its plot variability component (blue bars in Fig.3a). Accordingly, the model with the lowest MSS was model (21), i.e. the model with the lowest $(N/A) MSE_{\mathcal{F}}$. This selection agreed with the model selection based on tree-level performance statistics (Fig. 3a). For a plot area of 1 ha, MSS was still dominated by its plot variability component but the other components of MSS (violet and orange bars in Fig.3b) gained in importance. Model (21), which had a large prediction bias, was outperformed by model (24) that had the smallest prediction bias among the five models. For a large plot area of 10 ha, the plot variability component of MSS was no longer decisive in model selection (Fig. 3c). Thanks to its small prediction bias and coefficient variability, model (24) again outperformed the other models.

According to the F-test to compare nested models, which is a tree-level approach, model (23) outperformed model (22) ($F = 165.5$ with 841 and 840 df, p-value $< 0.001$). In other words, adding tree height on top of wood specific gravity and tree diameter in the model predictors improved the predictive performance of the model at the tree level. Whatever the plot area, the same conclusion was reached when comparing these two models at the plot level using MSS (compare models (22) and (23) in Fig.3). However, model comparison based on MSS did not always agree with the F-test. Model (21) outperformed model (20) according to the F-test ($F = 5.45$ with 842 and 746 df, p-value $< 0.001$). Based on MSS, the same conclusion was obtained for plot areas of 0.1 ha and 1 ha, but the performance ranking of the two models reversed for a plot area of 10 ha (compare models (20) and (21) in Fig.3). Model (24) did not outperform model (23) according to the F-test ($F = 0.20$ with 840 and 839 df, p-value = 0.66). In other words, adding the square log-diameter on top of wood specific gravity, diameter and height did not improve the predictive performance of the model at the tree level. Based on MSS, the same conclusion was obtained for a plot area of 0.1 ha, but the opposite conclusion was obtained for plot areas of 1 ha and 10 ha (compare models (23) and (24) in Fig. 3).

There is no F-test or goodness-of-fit statistic to compare a chain of models to a model. However, the MSS allowed us to compare the models to the two-step chain where tree height was first predicted from tree diameter, then biomass was predicted from wood specific gravity, diameter and height. Predicting tree height from diameter using a larger dataset reduced the prediction bias but brought some additional variability due to the coefficients of the height–diameter model. If we compare the chain to model (22) using MSS (compare (22) and (25) in Fig. 3), model (22) outperformed the chain for plot areas of 0.1 ha and 1 ha. However, as plot area increased and plot variability vanished, the chain became more performing than model (22).



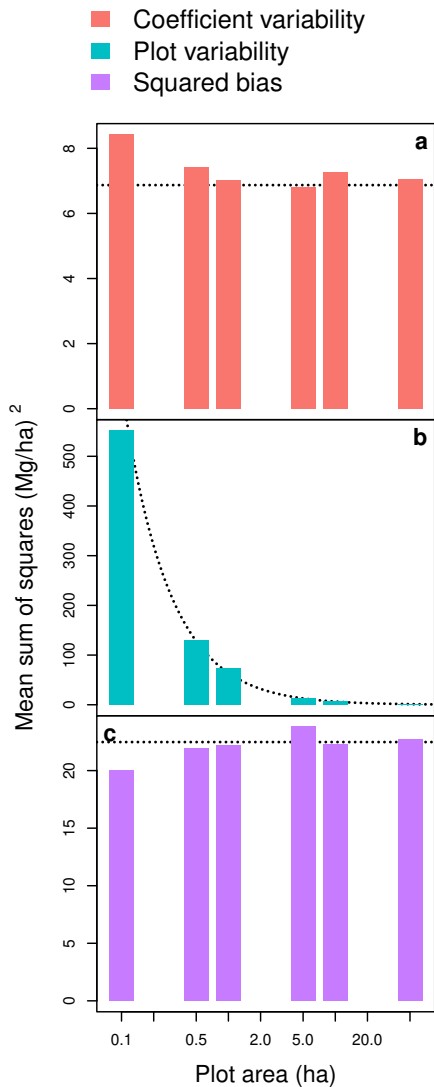

**Figure 2.** Coefficient variability (a), plot variability (b) and squared bias (c) as a function of plot area when predicting the aboveground biomass of a forest plot using the allometric equation: $\ln(B) = a_1 + b_1 \ln(\rho D^2 H)$ fitted to a dataset of 844 trees in the Congo basin. The dash lines are: (a) the horizontal line $y = N^2 \, \mathrm{ME}_{\mathcal{F}}$, (b) the line $y = (N/A) \, \mathrm{MSE}_{\mathcal{F}}$ where $A$ is the plot area, and (c) the horizontal line $y = (N\mathrm{b}_{\mathcal{F}})^2$, where forest plots are randomly generated according to a null forest model $\mathcal{F}$ with tree density $N$. $x$-axis has a log scale.





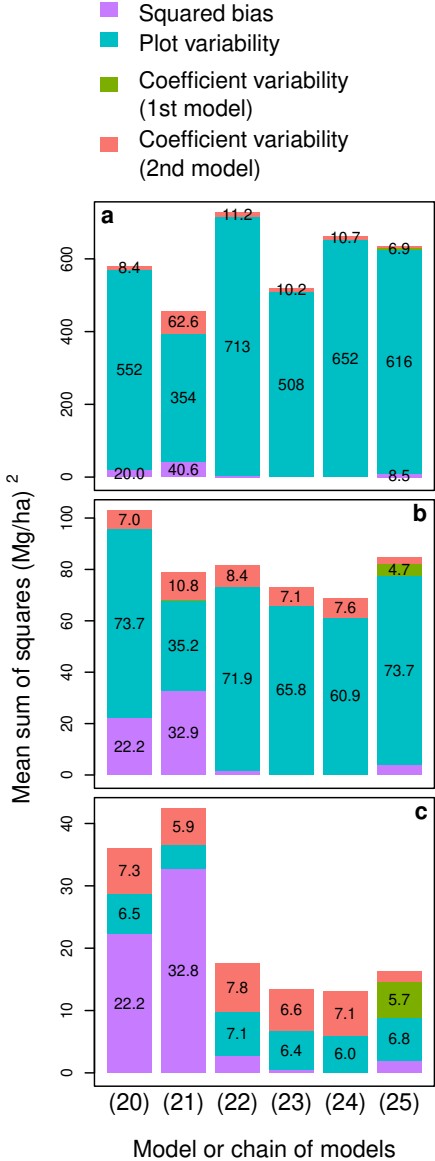

**Figure 3.** Partition of the mean sum of squared errors into squared bias, plot variability, the variability due the coefficients of the first model, and the variability due the coefficients of the second model, for six models or model chains (labeled on the $x$-axis) and for plots of area (a) $A = 0.1$ ha, (b) $A = 1$ ha, and (c) $A = 10$ ha. Errors are the differences between observed and predicted plot-level aboveground biomass. Model labels follow the model numbering in the text: (20): $\ln(B) = a_1 + b_1 \ln(\rho D^2 H)$; (21): $\ln(B) = a_{2s} + b_{2s} \ln(\rho D^2 H)$; (22): $\ln(B) = a_3 + b_3 \ln(\rho) + c_3 \ln(D)$; (23): $\ln(B) = a_4 + b_4 \ln(\rho) + c_4 \ln(D) + d_4 \ln(H)$; (24): $\ln(B) = a_5 + b_5 \ln(\rho) + c_5 \ln(D) + d_5 \ln(H) + e_5 [\ln(D)]^2$; (25): chain $(f \circ g)$ with $g$: $\ln(H) = a_6 + b_6 \ln(D)$ and $f$: $\ln(B) = a_1 + b_1 \ln(\rho D^2 H)$.



## 4   Discussion

### 4.1   Predictive performance statistics

Forest-level predictive performance statistics were quite different from tree-level ones. Model selection for allometric equations has been so far based on tree-level predictive performance such as AIC, residual standard error, or RMSE (e.g. Chave et al., 2014). Using forest-level performance statistics may thus shed a new light on the selection of allometric equations among competing models. The different weighing of trees in the dataset and in the null forest changed the performance statistics. For instance, large trees had a much stronger weight in the dataset $\mathcal{X}$ than in the null forest $\mathcal{F}$. Therefore, models that are biased

for large trees will counter-perform according to $b_{\mathcal{X}}$ but show better performance according to $b_{\mathcal{F}}$. This result also implies that different forests will give different performance statistics. In particular, the diameter distribution of trees in the forest will influence the forest-level performance statistics.

Model predictive performance at the tree level was not so different from the predictive performance at the plot level for the smallest plot size. For instance, the ranking of the five competing model based on their AIC (model $(21) > (23) > (24) > (20)$

$> (22)$) was almost the same as their ranking based on their MSS for a plot size of 0.1 ha (model $(21) > (23) > (20) > (24) > (22)$).

However, a significant result of our study was that the performance of a model to predict the biomass of a plot depended on plot size. This result is consistent with previous results based on error propagation (Chave et al., 2004). The plot-level performance statistics were good proxies of the MSS partition. Rather than doing long Monte Carlo computations to obtain

MSS, one can immediately approximate it as: $(Nb_{\mathcal{F}})^2 + (N/A)\,\mathrm{MSE}_{\mathcal{F}} + N^2\,\mathrm{ME}_{\mathcal{F}}$. This formula readily explains the change of MSS with plot size. For small plots, the predictive performance according to MSS is determined by the forest-level mean square error $\mathrm{MSE}_{\mathcal{F}}$. For large plots, individual tree errors compensate each other and cancel off, so that $\mathrm{MSE}_{\mathcal{F}}$ no longer matters for the predictive performance. The predictive performance is then determined by the prediction bias and the variability due to model coefficients. In other words, models with high residual standard error are strongly penalized in tree-level selection,

whereas it is much less a selection criterion in plot-level selection for large plots. When predicting the biomass of a large plot, what matters is the prediction bias and the coefficient variability.

Prediction bias and the mean square error (MSE), or related quantities such as relative bias, absolute bias, relative square error, etc., are often found in studies on allometric equations to report on the predictive performance of the models (Chave et al., 2014; Fayolle et al., 2018). In contrast, statistics on the variability due to the model coefficients are rarely found. Most

of the time, this variability is graphically shown as a confidence interval around the biomass prediction on a plot.

When developing allometric equations, a recurring question is whether it is worth adding a variable among the set of predictors of a model (Feldpausch et al., 2012; Goodman et al., 2014). This question is equivalent with comparing two nested models, one with the variable among its predictors and the other without. When there was strong indication that adding the variable improved the prediction of the biomass of trees, the same conclusion was reached when considering the biomass of

plots. For instance, adding tree height on top of diameter and wood density as a predictor of tree biomass strongly improved biomass predictions, whether at tree or at plot level. However, when the benefit from adding the variable was not so marked,





the conclusion based on tree-level prediction could differ from that based on plot-level prediction. All sorts of disagreement could be found. Tree-level prediction could be improved by adding the variable while plot-level prediction was not (which was here the case of the variable 'genus'). Or, on the contrary, tree-level prediction was not improved by adding the variable while
plot-level prediction was (which was here the case of the variable $\log(D)^2$).

### 4.2 Validation datasets

We made this study in the context of model fitting, i.e. when a calibration dataset is available and model coefficients need to be estimated. A different context is when models are given with know coefficients and a validation dataset is available to compare their predictive performance. The MSS computations can be done in both contexts. Nonetheless, for a calibration
dataset, by construction, model residuals sum up to zero. This property ensures that there is no prediction bias, at least for log-transformed variables and with equal tree weights in the dataset. This is no longer true with a validation dataset. If some model predictors vary in a systematic way (i.e. non-randomly) across plots in the validation dataset, then the model residuals will result in a plot-level prediction bias. To illustrate this effect, consider a pseudo-validation dataset $\mathcal{V}$ that is resampled from $\mathcal{X}$ using weights $w_i$ given by Eq. (1) but with the additional condition that a tree is sampled if its height is greater than the
nine tenths of the average height predicted by model (25). Dataset $\mathcal{V}$ is not a true validation dataset because it is built from the calibration dataset $\mathcal{X}$. Nonetheless, it illustrates what would happen if a validation dataset was taken from a plot where trees were systematically more slender than in the calibration plots. Then, the squared bias component of the MSS is indeed inflated. For model (20), it increases from 22 Mg$^2$/ha$^2$ to 130 Mg$^2$/ha$^2$ (Fig. 4, to be compared to model (20) in Fig. 3). In conclusion, the predictive performances of models can also be compared using MSS and a validation dataset, but it is likely to result in a
different model ranking than with the calibration dataset.

### 4.3 Local specific versus general equations

Our results can contribute to the long-standing debate about locally-developed specific equations versus general allometric equations (Chave et al., 2004; Weiskittel et al., 2015). Given a maximum sampling effort, and thus a given amount of available observations, the question is whether observations should be split among different categories (typically species and sites) to
305 fit locally-developed specific equations, or whether observations should be kept together to fit a general equation. Locally-developed specific equations tend to be less biased (Weiskittel et al., 2015). However, because they are based on a smaller number of observations, they tend to have a greater residual error and a greater variability due to coefficients. Our results indicate that the answer to this question also depends on the size of the plots for which biomass is predicted. Larger plots penalize biased models more heavily. They thus tend to favor locally-developed specific equations. Our results also show that
up to 1 ha of plot area, the plot variability is the dominant component of MSS. Because this component of MSS is the one related to the residual model error, it indicates that general allometric equations would tend be preferred to predict the biomass of plots with an area less than or equal to 1 ha. This conclusion would have to be confirmed using other datasets.

A variant of this question is whether to add an extra variable (typically tree height) as a predictor of the model or not (Feldpausch et al., 2012). The extra variable can better account for local variation in biomass, thus reducing bias. On the other



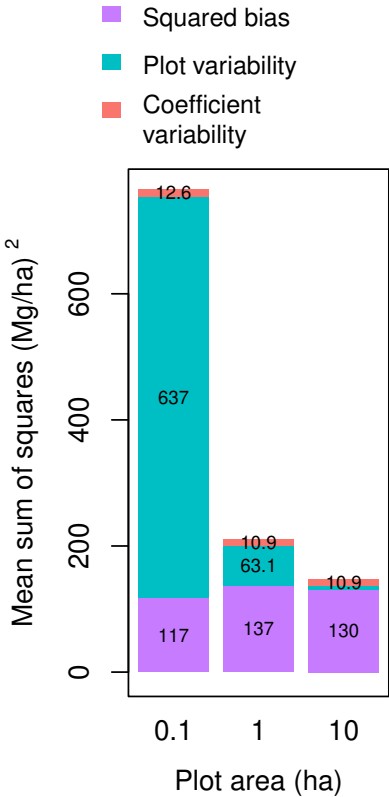

**Figure 4.** Partition of the mean sum of squared errors into squared bias, plot variability, and the variability due the model coefficients, for model $\ln(B) = a_1 + b_1 \ln(\rho D^2 H)$ and for plots of area 0.1 ha, 1 ha and 10 ha (on the $x$-axis). Plots are generated by keeping the slenderest trees in the dataset (i.e. excluding trees whose height is less than $0.9 \times$ average tree height). Errors are the differences between observed and predicted plot-level aboveground biomass.

hand, the requirement for this variable may reduce the availability of data, thus increasing residual error and the variability due to coefficients. Using a chain of models where the extra variable is first predicted from other predictors (typically tree height predicted from diameter) can circumvent this problem (Chave et al., 2014; Sullivan et al., 2018). Typically, the first model (the height–diameter relationship) is locally fitted while the second model (the biomass equation) is a general equation, thus combining the advantages of locally-fitted models with those of general models. We here showed that this strategy could

indeed be efficient, even for a single calibration dataset.

## 5   Conclusions

The plot-level predictive performance of an allometric equation depended on plot size. The effect of plot size $A$ could be well approximated by the formula: $(N b_{\mathcal{F}})^2 + (N/A) \mathrm{MSE}_{\mathcal{F}} + N^2 \mathrm{ME}_{\mathcal{F}}$, where the first term correspond to bias, the second



to the tree residual error, and the third one to the uncertainty on model coefficients. For small plots ($\leq 0.1$ ha), the plot-level
predictive performance was dominated by the $\mathrm{MSE}_{\mathcal{F}}$ term. Model selection based on plot-level predictive performance was
then consistent with model selection based on tree-level performance. For large plots, the term depending on $\mathrm{MSE}_{\mathcal{F}}$ vanished.
Model selection based on plot-level performance could then differ from that based on tree-level performance. In the case of
large plots, chains of models that combined a general equation to predict biomass and local equations to predicts some of the
predictors of the biomass equation could provide a good trade-off between the bias and the uncertainty on model coefficients.

*Code availability.* Code has been uploaded to Zenodo. https://doi.org/10.5281/zenodo.12748213

*Data availability.* We downloaded the data of Chave et al. (2014) at https://chave.ups-tlse.fr/pantropical_allometry.htm. The PREREDD+
data will be shared on reasonable request to the last author.

*Author contributions.* NP and AN conceived the ideas; NF, FBB, AF, JL, GNA, BS, ODYB, HMM and AN collected the data; NP designed
methodology, analysed the data and led the writing of the manuscript. All authors contributed critically to the drafts and gave final approval
for publication.

*Competing interests.* The authors declare no conflict of interests.

*Acknowledgements.* This work was supported by the PREREDD+ regional project, funded by a gift from the Global Environment Facility
[TF010038], administered by the World Bank to the COMIFAC. The component 2b of the PREREDD+, which aimed at 'Building allometric
equations for the forests of the Congo basin,' was led by the ONFi/TEREA/Nature+ consortium. Data collection in Gabon was carried out
by IRET with logistic support necessary for the field and laboratory measurements provided by Rougier Gabon.





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
