# Peer review of "Selecting allometric equations to estimate forest biomass from plotrather than individual-level predictive performance"

_EGUsphere, 2024_

## Referee Comment (RC1)

Review Egusphefe - **Selecting allometric equations to estimate forest biomass from plot-rather than individual-level predictive performance**

The paper, **Selecting allometric equations to estimate forest biomass from plot-rather than individual-level predictive performance**, makes significant and relevant contributions to forest biomass estimation, with outstanding strengths. For example, it proposes a robust and innovative method that focuses on the predictive performance of allometric models at the plot level rather than at the individual tree level. This is especially relevant because, in many cases, allometric models are primarily used to estimate biomass in forest plots rather than individual trees. The developed methodology considers that residual errors in individual trees can cancel each other out within a plot, which can affect the accuracy in selecting the most appropriate models. The study proposes a three-term formula that balances bias, residual error, and uncertainty in model coefficients, depending on the plot size. Overall, it is also observed that the novel results highlight that, for small plots, model selection based on tree-level and plot-level performance is consistent. However, for larger plots, this consistency disappears, suggesting that different selection criteria should be applied depending on the scale of the analysis. By offering a more precise approach to selecting allometric equations, the study contributes to improving forest biomass estimates, which is crucial for sustainable forest management and monitoring carbon stocks in the context of climate change. These strengths make the article relevant to the literature, offering practical and methodological solutions to improve the accuracy of biomass estimation in different forest contexts.

In summary, I have no comments that discredit the quality of this manuscript. However, I would like to read the authors' responses to the following questions:

Methodology:

1) Although the study focuses on the giant Congo rainforest, the study employs a detailed approach to estimate biomass in tropical forests using different plot size strategies. Is the methodology used to calibrate and validate the models robust enough for possible different types of tropical forests? How do these methodologies deal with the heterogeneity of tropical forests, which can vary significantly in terms of structure and species composition?

2) Considering that field data collection is essential for model calibration, how was the potential bias from limited or non-representative sampling of different forest areas addressed? Can this be addressed in the manuscript?

Results, broader implications, and limitations of the study:

1) How do the authors interpret the results found regarding spatial and temporal variability of biomass in the studied forests? Is there any indication of changes in biomass stock over time that could be correlated with environmental or anthropogenic factors?

2) Is biomass quantification in line with estimates from similar studies? Can you provide data showing or not showing discrepancies, and what might explain them?

3) The results indicate that tropical forests have a significant capacity to store biomass. What are the implications of these findings for conservation and climate change mitigation policies? How do these results contribute to the global understanding of the role of tropical forests as carbon sinks?

4) To what extent does this study advance knowledge on the quantification of biomass and carbon in tropical forests? How does it contribute to the development of new methodologies or the improvement of existing methodologies?

5) How can the results of this study influence future research on changes in carbon stocks in tropical forests? Are there gaps that still need to be addressed?

6) What is the impact of this study on understanding the role of tropical forests in carbon sequestration, especially in the context of global climate change?

7) What are the main limitations of the methods used in the study, especially in terms of spatial scale? How might these limitations affect the interpretation of the results?

8) What were the main challenges in quantifying uncertainty associated with the methods associated with different plot sizes and individual trees, and how might this influence the results?

9) Are there any limitations related to the representativeness of the field data about the diversity of tropical forests? How might the lack of representativeness have impacted the results?

Main Scientific Contributions

- The study offers significant contributions to the science of ecology and the understanding of tropical forests as carbon sinks. What are the main methodological innovations presented?

- How does the study advance knowledge about spatial variation of biomass in tropical forests? What new insights does it offer for these forests' conservation and sustainable management?

- How can the results of this study be applied to other tropical regions in addition to the areas studied? Is there potential for replicating the methodologies in other areas or biomes?

I believe that the authors' input on these questions is crucial and that it can significantly contribute to the ongoing discussion of the manuscript. Their responses can guide a critical evaluation of the article, highlighting both its contributions and the areas that can be improved or better explored in future research.